# Regularising for invariance to data augmentation improves supervised learning

## Abstract

Data augmentation is used in machine learning to make the classifier invariant to label-preserving transformations. Usually this invariance is only encouraged implicitly by sampling a single augmentation per image and training epoch. However, several works have recently shown that using multiple augmentations per input can improve generalisation or can be used to incorporate invariances more explicitly. In this work, we first empirically compare these recently proposed objectives that differ in whether they rely on explicit or implicit regularisation and at what level of the predictor they encode the invariances. We show that the predictions of the best performing method are also the most similar when compared on different augmentations of the same input. Inspired by this observation, we propose an explicit regulariser that encourages this invariance on the level of individual model predictions. Through extensive experiments on CIFAR-100 and ImageNet we show that this explicit regulariser (i) improves generalisation and (ii) equalises performance differences between all considered objectives. Our results suggest that objectives that encourage invariance on the level of the neural network features itself generalise better than those that only achieve invariance by averaging predictions of non-invariant models.

## 1 Introduction

In supervised learning problems, we often have prior knowledge that the labels should be invariant or insensitive to certain transformations of the inputs; though it is often difficult to make the classifier explicitly invariant to all valid input transformations in a tractable way (Niyogi et al., 1998). Data augmentation (DA) is a widely used technique to incorporate such an inductive bias into the learning problem implicitly: it enlarges the training set with randomly transformed copies of the original data to encourage the classifier to be correct on larger parts of the input space.

When using DA, practitioners typically sample a single augmentation per image and minibatch during training. However, Fort et al. (2021) recently showed that this can introduce a detrimental variance that slows down training. Instead, using *several* augmentations per image in the same minibatch can reduce this variance and improve the classifier's generalisation performance by leveraging the useful bias of DAs more effectively (Fort et al., 2021; Hendrycks et al., 2020; Hoffer et al., 2020; Touvron et al., 2021). This modified objective simply takes the average of the individual losses over different augmentations of the same input, which encourages the model function to produce similar predictions for every augmentation.

The choice of averaging the individual losses motivates the question of whether there are better ways of combining the model outputs when sampling multiple augmentations per input. In this direction, Nabarro et al. (2021) recently studied principled ways to incorporate data-augmentation in a Bayesian framework for neural networks by explicitly constraining the classifier outputs to be invariant to DA by averaging either (i) the post-softmax probabilities or (ii) the pre-softmax logits during training. This alternative of averaging the probabilities is often used to improve test performance by making ensemble predictions over different augmentations of the test input (Krizhevsky et al., 2012; Simonyan and Zisserman, 2014; Szegedy et al., 2015). Interestingly, while Nabarro et al. (2021) showed that both Bayesian-inspired approaches improved performance compared to sampling a single augmentation per input, they do not compare them to the

**(a) Data flow with multiple augmentations**  **(b) Objectives using multiple augmentations**  **(c) KL regulariser**

**Figure 1:** We investigate three objectives that use multiple augmentations per input $\boldsymbol{x}$: $\mathcal{L}_{\langle\text{losses}\rangle}$ averages the losses of individual augmentations, while $\mathcal{L}_{\langle\text{probs}\rangle}$ and $\mathcal{L}_{\langle\text{logits}\rangle}$ average the probabilities and logits, respectively (*left* and *middle*). We further propose a regulariser that averages the KL-divergence between all pairs of predictive distributions over different augmentations of the same input (*right*).

baseline of averaging the losses, perhaps because this baseline does not correspond to a valid likelihood on the unaugmented dataset.

As a result, it remains an open question how and at what level practitioners should incorporate their prior knowledge of invariance in the model:

1. *implicitly* by averaging the losses over individual augmentations;

2. *explicitly* by constructing a model whose prediction is defined as the average over all augmentations, as is done when averaging the probabilities or the logits.

In this work we first compare these two alternative perspectives. We show that averaging the losses during training leads to better generalisation performance, even when ensembling predictions for different augmentations at test time. Additionally, we empirically show that the approach of averaging the losses makes the predictions of the neural network model for different augmentations significantly more similar than the other two methods. Based on this observation, we conjecture that having more invariant predictions for individual data augmentations is the reason why averaging the losses generalises better.

To further investigate this hypothesis, we introduce a regulariser that explicitly encourages individual predictions for different DAs of the same input to be similar. Since we already use multiple DAs per input in our supervised setup, we can directly compare the predictive distributions of different DAs at no additional computational cost. Through extensive experiments, we show that this explicit regulariser consistently improves generalisation by making the model more invariant to DAs of the same input. Furthermore, we show that using this regulariser equalises the performance differences between averaging the losses vs. averaging the logits or probabilities. These results corroborate our conjecture that encouraging invariance on the level of the network outputs is better than achieving invariance by averaging non-invariant models alone.

**To summarise, our main contributions are:**

1. We study the role of explicit and implicit invariance in supervised learning. We find that a naive approach of averaging the losses outperforms Bayesian-inspired losses, and results in more invariant model predictions (Sec. 3).

2. We propose a regulariser for supervised learning tasks that explicitly encourages predictions of different DAs for the same input to be similar to each other (Sec. 4).

3. We show that the regulariser improves generalisation for all base losses, equalising the performance differences between methods; this suggests that encouraging invariance on the level of the network outputs works better.

## 2  Background

In this work, we consider the supervised learning setting with input vectors $\boldsymbol{x} \in \mathcal{X} \subseteq \mathbb{R}^d$ and corresponding labels $y \in \mathcal{Y} = \{1, \ldots, C\}$. Let $\mathcal{D} = \{(\boldsymbol{x}_n, y_n)\}_n^N$ denote the training set and $\boldsymbol{f}_\phi$ be a parametric (neural network) model with parameters $\phi$. The standard objective is to minimise the empirical negative log likelihood of the training data:

$$\min_{\phi} \quad \mathbb{E}_{(\boldsymbol{x}, y) \sim \mathcal{D}} \left[ \mathcal{L}_\phi(\boldsymbol{x}, y) \right], \qquad \mathcal{L}_\phi(\boldsymbol{x}, y) = -\log p(y | g(\boldsymbol{f}_\phi(\boldsymbol{x}))). \tag{1}$$

Here, $g$ is an inverse link function (such as the logistic or softmax) that maps the output of the function $\boldsymbol{f}_\phi$ to a probability distribution over $\mathcal{Y}$, and $\mathcal{L}_\phi(\boldsymbol{x}, y)$ denotes the negative log likelihood objective for a single datapoint $(\boldsymbol{x}, y)$.

**Standard data augmentation (DA).**  DA refers to the practice of enlarging the original dataset $\mathcal{D}$ by applying label-preserving transformations to its inputs $\boldsymbol{x}$. The transformations are domain- and dataset-specific and hand-engineered to incorporate the right inductive biases such as invariances or symmetries (Niyogi et al., 1998). For example, in image classification, we commonly use a combination of discrete transformations such as horizontal flips and continuous transformations such as shears or changes in brightness or contrast (Cubuk et al., 2020). Following Wilk et al. (2018), we call the distribution over all augmentations $\widetilde{\boldsymbol{x}}$ for a given input $\boldsymbol{x}$ its *augmentation distribution* and denote it by $p(\widetilde{\boldsymbol{x}} | \boldsymbol{x})$. The per-input objective that corresponds to regular neural network training with DA is then given by:

$$\mathcal{L}_{\langle \text{losses} \rangle}(\boldsymbol{x}, y) = \mathbb{E}_{\widetilde{\boldsymbol{x}} \sim p(\widetilde{\boldsymbol{x}} | \boldsymbol{x})} \left[ \mathcal{L}_\phi(\widetilde{\boldsymbol{x}}, y) \right], \tag{2}$$

and can be interpreted as averaging the losses for different augmentations of the same input. The expectation in Eq. (2) is commonly approximated by a single Monte Carlo sample, i.e., for each input in a minibatch we sample one augmentation. Several recent works have shown, however, that sampling a larger number of augmentations per input in a minibatch can be beneficial (Fort et al., 2021; Hendrycks et al., 2020; Hoffer et al., 2020; Touvron et al., 2021). This phenomenon has been attributed to the observation that sampling multiple augmentations per input in a minibatch has the effect of reducing the variance arising from the DA procedure and empirically improves generalisation (Fort et al., 2021). This is in contrast to minibatch sampling where lowering the variance by increasing the batch size can reduce generalisation (Smith et al., 2020).

**Bayesian-inspired DA.**  The question of how best to incorporate DA in Bayesian deep learning has recently also received some attention, especially since the naive approach of enlarging the training set based on the number of augmentations results in overcounting the likelihood w.r.t. the prior (Nabarro et al., 2021). In this context, Wenzel et al. (2020) noted that the data augmented objective in Eq. (2) cannot be interpreted as a valid likelihood objective. Nabarro et al. (2021) argued that DA should be viewed as nuisance variables that should be marginalised over, and proposed two Bayesian-inspired objectives that construct an invariant predictive distribution by integrating a non-invariant predictor over the augmentation distribution, where either the post-softmax probabilities or pre-softmax logits are averaged:

$$\mathcal{L}_{\langle \text{probs} \rangle}(\boldsymbol{x}, y) = -\log \mathbb{E}_{\widetilde{\boldsymbol{x}} \sim p(\widetilde{\boldsymbol{x}} | \boldsymbol{x})} \left[ p(y | g(\boldsymbol{f}_\phi(\widetilde{\boldsymbol{x}}))) \right], \tag{3}$$

$$\mathcal{L}_{\langle \text{logits} \rangle}(\boldsymbol{x}, y) = -\log p(y | g(\mathbb{E}_{\widetilde{\boldsymbol{x}} \sim p(\widetilde{\boldsymbol{x}} | \boldsymbol{x})} \left[ \boldsymbol{f}_\phi(\widetilde{\boldsymbol{x}}) \right])). \tag{4}$$

This construction makes the predictive distribution *explicitly* invariant to DAs. This is in contrast to Eq. (2), for which invariance is only encouraged *implicitly* by minimising the loss for each augmentation independently.

**Finite sample DA objectives.**  In this work, we first compare the above two alternate perspectives on DA to understand how best to incorporate invariance into the model. Because exact marginalisation in Eqs. (2) to (4) is intractable, we approximate it with $K$ samples from the augmentation distribution giving rise to the following objectives:

Objectives using multiple augmentations

$$\widehat{\mathcal{L}}^K_{\langle\text{losses}\rangle}(\boldsymbol{x}, y) = \frac{1}{K} \sum_{k=1}^{K} \mathcal{L}\left(g(\boldsymbol{f}_\phi(\widetilde{\boldsymbol{x}}_k)), y\right) \tag{5}$$

$$\widehat{\mathcal{L}}^K_{\langle\text{probs}\rangle}(\boldsymbol{x}, y) = \mathcal{L}\left(\frac{1}{K} \sum_{k=1}^{K} g(\boldsymbol{f}_\phi(\widetilde{\boldsymbol{x}}_k)), y\right) \tag{6}$$

$$\widehat{\mathcal{L}}^K_{\langle\text{logits}\rangle}(\boldsymbol{x}, y) = \mathcal{L}\left(g\left(\frac{1}{K} \sum_{k=1}^{K} \boldsymbol{f}_\phi(\widetilde{\boldsymbol{x}}_k)\right), y\right) \tag{7}$$

Here the $\widetilde{\boldsymbol{x}}_k, k \in \{1, \dots, K\}$, are the $K$ augmentations for an input $\boldsymbol{x}$ independently sampled from $p(\widetilde{\boldsymbol{x}}|\boldsymbol{x})$. For $K = 1$ the losses are equivalent. In Fig. 1 (a) and (b) we illustrate these three objectives for $K = 3$. See App. A for a brief analysis of how these three losses compare to each other, and how the finite sample versions of the objectives (Eqs. (5) to (7)) relate to the original objectives (Eqs. (2) to (4)).

**Test-time DA (TTA).**  While so far we have discussed DA during *training*, it is also common to employ it at test time by making ensemble predictions over multiple augmentations of the test input to boost performance (Krizhevsky et al., 2012; Simonyan and Zisserman, 2014; Szegedy et al., 2015). Note that for TTA, practitioners typically *average the probabilities* as in Eq. (6), while for training practitioners typically *average the losses* as in Eq. (5). This inconsistency further motivates the question of whether it is also better to similarly ensemble the probabilities during training.

## 3 Experimentally comparing the two alternate perspectives on DA

In Sec. 2, we described two alternate perspectives on DA to incorporate invariance into the model: either implicitly by averaging the losses (Eq. (5)), or explicitly by averaging either the probabilities (Eq. (6)) or the logits (Eq. (7)). To better understand how these perspectives compare, in this section we perform a thorough experimental evaluation of the three objectives. Specifically, we compare both generalisation performance and the amount of invariance induced by the three losses for varying number of augmentations per input per minibatch during training (which we call *augmentation multiplicity* $K_{\text{train}}$). This evaluation extends previous limited results by Nabarro et al. (2021), who only compare $\widehat{\mathcal{L}}^K_{\langle\text{probs}\rangle}$ and $\widehat{\mathcal{L}}^K_{\langle\text{logits}\rangle}$ for $K_{\text{train}} \in [1, 6]$ and do not investigate $\widehat{\mathcal{L}}^K_{\langle\text{losses}\rangle}$ for $K_{\text{train}}$ larger than 1, and Fort et al. (2021), who investigate $\widehat{\mathcal{L}}^K_{\langle\text{losses}\rangle}$ but do not consider invariances.

**Experimental setup.**  Since the performance of models with batch normalisation depends strongly on the examples used to estimate the batch statistics (Hoffer et al., 2017), in the main paper we train on highly performant models that do not use batch normalisation, following Fort et al. (2021), to simplify our analysis (see App. D.5 for more discussion). We use the following networks: a WideResNet 16-4 (Zagoruyko and Komodakis, 2016) with SkipInit initialisation (De and Smith, 2020) for CIFAR-100 classification, and an NF-ResNet-101 (Brock et al., 2021a) for ImageNet classification. For both datasets we use standard random crops and random horizontal flips for DA following previous work (Brock et al., 2021a; Zagoruyko and Komodakis, 2016). See Sec. 5.1 for additional experiments with a wider range of data augmentations where we have similar findings.

We use augmentation multiplicities $K_{\text{train}} = 1, 2, 4, 8, 16$ for CIFAR-100 and $K_{\text{train}} = 1, 2, 4, 8$ for ImageNet due to computational constraints. We also fix the total batch size, which implies that for $K_{\text{train}} > 1$ the number of unique images per single batch decreases proportionally and the total number of parameter updates for the same epoch budget increases (Fort et al., 2021). Because the optimal epoch budget might change with the augmentation multiplicity (Fort et al., 2021), for each experiment we run an extensive grid-search for the optimal learning rate and optimal epoch budget for every value of $K_{\text{train}}$ for the three objective functions. For evaluation, we compute the top-1 test accuracy both using standard central-crops as well as using test time augmentations (TTAs) with the number of augmentations set to $K_{\text{test}} = 16$ for CIFAR-100 and $K_{\text{test}} = 8$ for ImageNet. As a measure of the invariance of the predictions, we calculate the KL divergence between the predictive distributions of different augmentations of the same input (see Sec. 4 for a more detailed description of this measure). We run each CIFAR-100 experiment 5 times with different random seeds; for the ImageNet experiments we only use a single seed due to computational constraints. For further details on the experimental setup, please refer to App. C .

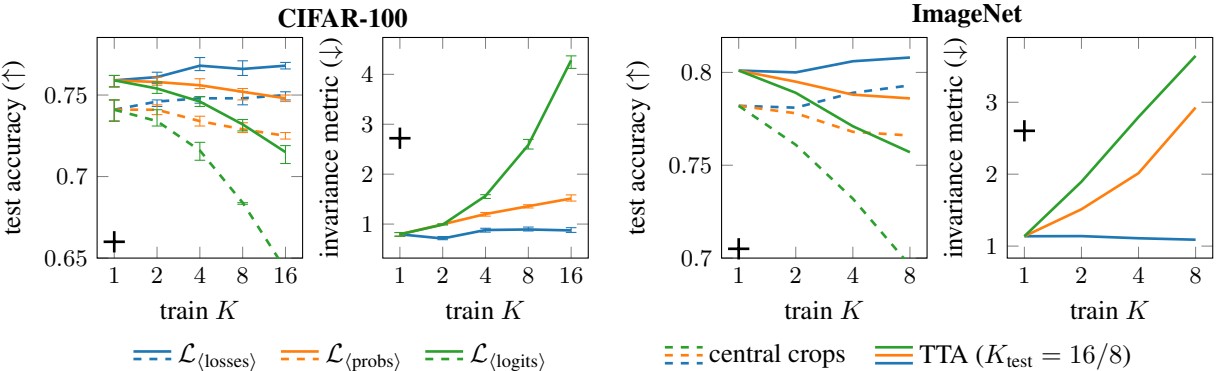

**Figure 2:** The three losses differ in their generalisation performance *(left)* as well as in how invariant the predictions are w.r.t. data augmentations *(right)*. For reference, $+$ mark models trained without DA (Eq. (1)).

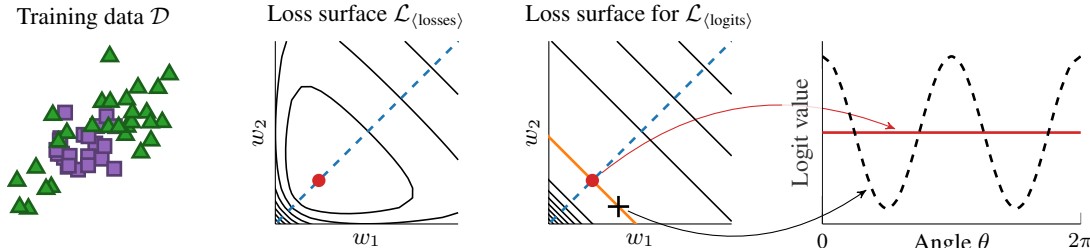

**Figure 3:** Example why $\mathcal{L}_{\langle\text{losses}\rangle}$ encourages invariance but $\mathcal{L}_{\langle\text{logits}\rangle}$ does not. We fit a 2-parameter model to a binary classification problem with rotational symmetry *(left)*. Setting $w_1 = w_2$ (- - -) corresponds to models with constant logit values regardless of the angle $\theta$. The loss surface for $\mathcal{L}_{\langle\text{losses}\rangle}$ has a single optimum at an invariant model (•), whereas the loss surface for $\mathcal{L}_{\langle\text{logits}\rangle}$ is optimal anywhere along a line ($\searrow$), as it only constrains the *average* logit value. The model is invariant only for one point along the line (•), whereas for all other values (e.g. $+$) the logit value changes with the angle $\theta$.

**Experimental results.** Overall, the results are qualitatively the same on both datasets. For top-1 test accuracy we find that (see Fig. 2 *left* and Tab. 1): (i) Using augmentations at test time is consistently at least $\sim 2\%$ better than predictions on central crops; (ii) Averaging the losses ($\mathcal{L}_{\langle\text{losses}\rangle}$) clearly performs better than averaging the probabilities ($\mathcal{L}_{\langle\text{probs}\rangle}$) which performs better than averaging the logits ($\mathcal{L}_{\langle\text{logits}\rangle}$); (iii) While $\mathcal{L}_{\langle\text{losses}\rangle}$ improves with larger augmentation multiplicity as reported by Fort et al. (2021), both Bayesian-inspired objectives, $\mathcal{L}_{\langle\text{probs}\rangle}$ and $\mathcal{L}_{\langle\text{logits}\rangle}$, consistently *degrade* in performance as $K_{\text{train}}$ increases.

When comparing how the invariance measure for different augmentations of the same input changes as we vary the augmentation multiplicity, we find that (see Fig. 2 *right*), while it stays relatively constant when averaging the losses, it markedly increases for both Bayesian-inspired losses as $K_{\text{train}}$ increases. We therefore conjecture that: the beneficial bias of DA, as discussed by Fort et al. (2021), is that it promotes invariance of the model to DA; the detrimental variance also introduced by DA can be mitigated by using larger augmentation multiplicities. This also explains why larger $K_{\text{train}}$ do not further reduce the invariance measure for $\mathcal{L}_{\langle\text{losses}\rangle}$. For completeness, we also show in Fig. 2 that a network trained *without* DA has a much higher invariance measure than networks trained with DA when $K_{\text{train}} = 1$.

Perhaps a somewhat counter-intuitive result from Fig. 2 is that the Bayesian-inspired losses lead to worse performance and less invariant predictions as we increase $K_{\text{train}}$.[1] We make two arguments to explain this observation.

---

[1] This observation contradicts results by Nabarro et al. (2021). While we were able to reproduce their results using their model (a ResNet18 that uses batch normalisation), their observations do not extend to any of our normaliser-free models. For completeness we include a comparison on their model in App. D.5

| Dataset | Network | KL regulariser | $K_{\text{train}} = 1$ | $K_{\text{train}} = 8$ or 16 | | |
|---------|---------|----------------|------------------------|------------------------------|---|---|
| | | | | $\mathcal{L}_{\langle\text{losses}\rangle}$ | $\mathcal{L}_{\langle\text{probs}\rangle}$ | $\mathcal{L}_{\langle\text{logits}\rangle}$ |
| CIFAR-100 | WRN 16-4 | ✗ | $0.741_{\pm 0.003}$ | $0.750_{\pm 0.001}$ | $0.725_{\pm 0.001}$ | $0.643_{\pm 0.001}$ |
| CIFAR-100 | WRN 16-4 | ✓ | $0.741_{\pm 0.003}$ | $0.759_{\pm 0.001}$ | $0.758_{\pm 0.002}$ | $0.759_{\pm 0.001}$ |
| ImageNet | NF-ResNet-101 | ✗ | 0.782 | 0.793 | 0.766 | 0.696 |
| ImageNet | NF-ResNet-101 | ✓ | 0.782 | 0.801 | 0.804 | 0.804 |
| ImageNet | NFNet-F0 | ✗ | 0.803 | 0.813 | 0.788 | 0.717 |
| ImageNet | NFNet-F0 | ✓ | 0.803 | 0.819 | 0.822 | 0.819 |

**Table 1:** Top-1 test accuracy generalisation performance for all three losses with and without regulariser evaluated on central crops. For the losses we use $K_{\text{train}} = 16$ on CIFAR and $K_{\text{train}} = 8$ on ImageNet. The NFNet-F0 uses RandAugment as DA in addition to the horizontal flips and random crops used for the WRN and NF-ResNet. For further results, please see App. D.

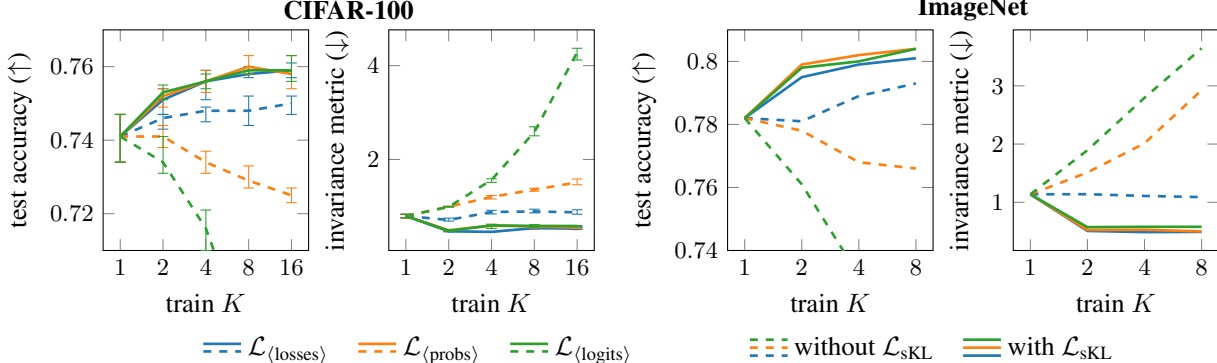

**Figure 4:** Adding the proposed KL regulariser $\mathcal{L}_{\text{sKL}}$ makes the individual predictions more invariant to DA and further boosts generalisation performance (shown here: top-1 test accuracy on central crops).

First, we illustrate in an example that the Bayesian-inspired objectives can easily have many non-invariant solutions to simple problems with symmetries compared to $\mathcal{L}_{\langle\text{losses}\rangle}$. We consider a simple 2-dimensional binary classification problem with azimuthal (rotational) symmetry in Fig. 3 and use a linear 2-parameter model that is rotationally invariant when its parameters are equal. The loss surface for $\mathcal{L}_{\langle\text{losses}\rangle}$ only has a single minimum, which is invariant. However, since $\mathcal{L}_{\langle\text{logits}\rangle}$ only constrains the *average* logit value, the loss surface for $\mathcal{L}_{\langle\text{logits}\rangle}$ is also minimised anywhere along a line of non-invariant models, for which the individual logit values vary with the angle (DA).

Second, we note that when we average the losses we make a prediction with every one of the $K_{\text{train}}$ augmentations and every one of them has to explain the (same) label, thus encouraging the predictions to be similar. In contrast, for $\mathcal{L}_{\langle\text{logits}\rangle}$ (or $\mathcal{L}_{\langle\text{probs}\rangle}$) we only make a single prediction using the average of the logits (or probabilities) of the individual augmentations. How similar or different the individual values are is irrelevant as only their average matters. We speculate that as $K_{\text{train}}$ grows, it is sufficient for some of the augmentations to have confident enough predictions that dominate the average, such that there is very little pressure for some of wrong or less confident augmentations to explain the data well, and this pressure might only decrease as $K_{\text{train}}$ increases.

## 4    A KL-regularised objective leads to more invariance and better generalisation

In Sec. 3 we showed that averaging the losses over DAs results in better generalisation compared to averaging the logits or probabilities. We hypothesised that this improvement in performance was due to the more invariant individual predictions for $\mathcal{L}_{\langle\text{losses}\rangle}$. To further investigate this hypothesis, we propose to explicitly

regularise the parametric model $\boldsymbol{f}_\phi$ to make more similar predictions across individual DAs of the same input. We then study the effect of this invariance regulariser on the generalisation performance of the model.

While the classification losses achieve some degree of invariance, we wish to encourage this objective more strongly and directly. Since the ultimate quantity of interest in supervised learning is the predictive distribution over labels, $p(y|g(\boldsymbol{f}_\phi(\boldsymbol{x})))$, a natural choice for such a regulariser is to penalise the KL divergence between predictives for different DAs. This is in contrast to many self-supervised methods that penalise differences between vectors in an arbitrarily chosen embedding space (Chen et al., 2020b). The KL divergence is attractive since it regularises the entire distribution and not just incorrect predictions. Moreover, it is an information theoretic quantity measured in nats like the original log-likelihood loss (Eq. (1)), which makes it easier to compare their values and reason about the regularisation strength.

Since the KL divergence is asymmetric, we use its symmetrised version, also referred to as Jeffrey's divergence:

$$\mathcal{L}_{\text{sKL}}(\boldsymbol{x}) = \mathbb{E}_{\widetilde{\boldsymbol{x}}, \widetilde{\boldsymbol{x}}' \sim p(\widetilde{\boldsymbol{x}}|\boldsymbol{x})} \left[ \text{KL}(p(y|\widetilde{\boldsymbol{x}}) \parallel p(y|\widetilde{\boldsymbol{x}}')) \right]. \tag{8}$$

We add $\mathcal{L}_{\text{sKL}}$ as a soft-constraint regulariser to the original objectives Eqs. (2) to (4) with a regularisation strength $\lambda$:

$$\mathcal{L}_{\langle \dots \rangle, \text{ regularised}}(\boldsymbol{x}, y) = \mathcal{L}_{\langle \dots \rangle}(\boldsymbol{x}, y) + \lambda \mathcal{L}_{\text{sKL}}(\boldsymbol{x}). \tag{9}$$

We found that $\lambda = 1$ works well in practice and fix $\lambda$ to this value. Since the loss and the regulariser are on the same scale, this is a natural choice. We discuss this further in Sec. 5.3. We also emphasise that we define the regulariser using the predictive distributions of *individual* augmentations even though $\mathcal{L}_{\langle \text{probs} \rangle}$ and $\mathcal{L}_{\langle \text{logits} \rangle}$ only make a single prediction using *average* probabilities or logits, respectively.

Similar KL divergence-based regularisers have been explored as objectives in the contrastive and self-supervised learning literature recently; for example, Xie et al. (2020) use a cross-entropy to regularise predictions on unlabelled data in semi-supervised learning, while Mitrovic et al. (2021) target the predictive distribution of surrogate task-labels in fully unsupervised learning.

Finally, we note that the symmetrised KL is only one choice to measure invariance. We also consider evaluations using the L2 distance and cosine similarity between logits of augmentated images in App. D.1 and Fig. 8 and find that all three metrics qualitatively agree.

In practice we replace the expectation in Eq. (8) by a Monte Carlo estimate $\widehat{\mathcal{L}}_{\text{sKL}}^K$ with size determined by the augmentation multiplicity $K$ used to evaluate the main objective:

---
**Invariance regulariser used as soft-constraint**

$$\widehat{\mathcal{L}}_{\text{sKL}}^K(\boldsymbol{x}) = \frac{1}{K^2 - K} \sum_{\substack{k, k'=1 \\ k \neq k'}}^{K} \text{KL}(p(y|\widetilde{\boldsymbol{x}}_k) \parallel p(y|\widetilde{\boldsymbol{x}}_{k'})). \tag{10}$$
---

Eq. (10) is an unbiased estimate of Eq. (8) (see App. A for the derivation), and we illustrate it for $K = 3$ in Fig. 1 (c).

## 4.1 Experimental evaluation

To evaluate the effect of the KL regulariser, we add it to the three objectives discussed in Sec. 3 and run otherwise identical experiments on the WideResNet 16-4 with SkipInit on CIFAR-100 and the NF-ResNet-101 on ImageNet. See App. C for more experimental details.

The qualitative results again agree on both experiments (see Fig. 4 and Tab. 1). We find that the regularised objectives consistently perform better than their non-regularised counterparts. Perhaps more interestingly, all three regularised objectives now generalise equally well. Furthermore, we find that the invariance measure now is almost identical for all three regularised objectives as we vary the train augmentation multiplicity.

These results support our conjecture that the main driver of generalisation performance when using multiple DAs is the invariance of the *individual* predictions; simply constructing an invariant predictor through

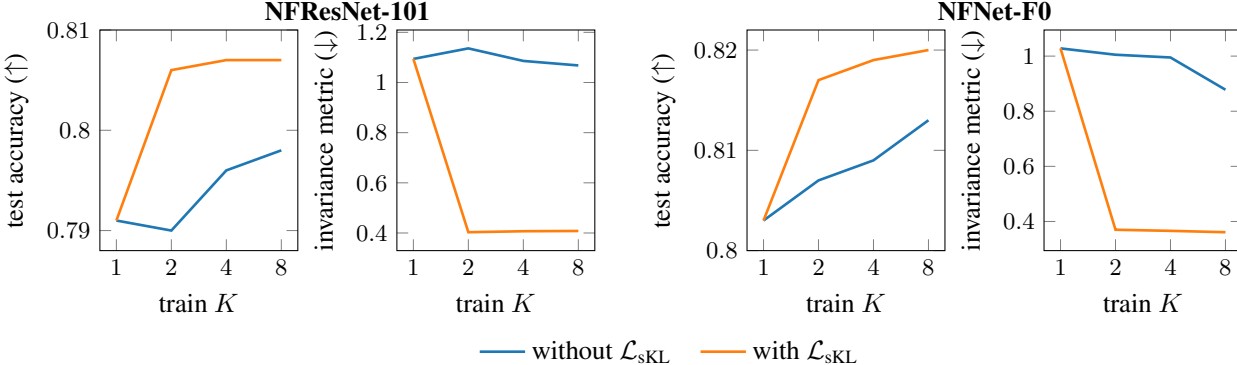

**Figure 5:** Networks trained on ImageNet with $\mathcal{L}_{\langle\text{losses}\rangle}$ and stronger data augmentations (RandAugment).

averaging non-invariant features, as is done in the Bayesian-inspired losses, is not sufficient . When we account for this by adding our proposed regulariser, which further encourages invariance on the level of individual predictions, all objectives improve and now show similar performance.

## 5 Ablations

In this section, we provide additional experiments that support our main findings. In particular, we show that our results also hold for a much wider range of data augmentations, and when changing the capacity of our models. We also study the effect of varying the strength $\lambda$ of the KL regulariser. In addition we include extended experimental results in App. D: We provide further ablations on using alternative invariance measures for evaluation (App. D.1), study the influence of the augmentation multiplicity at test time (Apps. D.2 and D.3), consider networks *with* BatchNorm (Apps. D.4 and D.5), and provide analyses on further variants of our BatchNorm-free network architectures (App. D.6). They are in line with the presented results in the main paper.

### 5.1 Using a larger set of data augmentations

In all previous experiments we only used horizontal flips and random crops as DAs. In this section, we show that our main findings also apply more generally when using stronger DAs and for more modern image classification architectures, namely NFNets (Brock et al., 2021b). As stronger DAs, we consider RandAugment (Cubuk et al., 2020) in addition to horizontal flips and random crops, which use a combination of 16 augmentations such as colour and brightness changes or image rotations, shears, and distortions. We set the magnitude of RandAugment to 5 following the setting used for the NFNet-F0 in Brock et al. (2021b).

We first train the NF-ResNet-101 on ImageNet with the same experimental setup as before but adding RandAugment to the set of DAs. We then also train an NFNet-F0 (Brock et al., 2021b) on ImageNet; the NFNet models are highly expressive, and therefore prone to overfit, and rely on strong DAs to achieve good generalisation performance (Brock et al., 2021b). We exclude any DAs that use mixing between different inputs, such as CutMix (Yun et al., 2019) and Mixup (Zhang et al., 2018) such that the model only uses RandAugment, horizontal flips and random crops as DA. Due to computational constraints, we only run experiments with the $\mathcal{L}_{\langle\text{losses}\rangle}$ objective and its regularised version for different augmentation multiplicities. We also modify the training procedure to make the setting more comparable to our other experiments; see App. C for details. While these modifications slightly reduce performance of the NFNet-F0, the baseline model ($K_{\text{train}} = 1$) still achieves 80.3% top-1 accuracy on central crops compared to 83.6% as reported by Brock et al. (2021b).

For both experiments, the results shown in Fig. 5 qualitatively agree with those discussed in Sec. 4. In particular, we see that the KL regulariser improves performance over the non-regularised counterpart, and again results in significantly more invariant predictions.

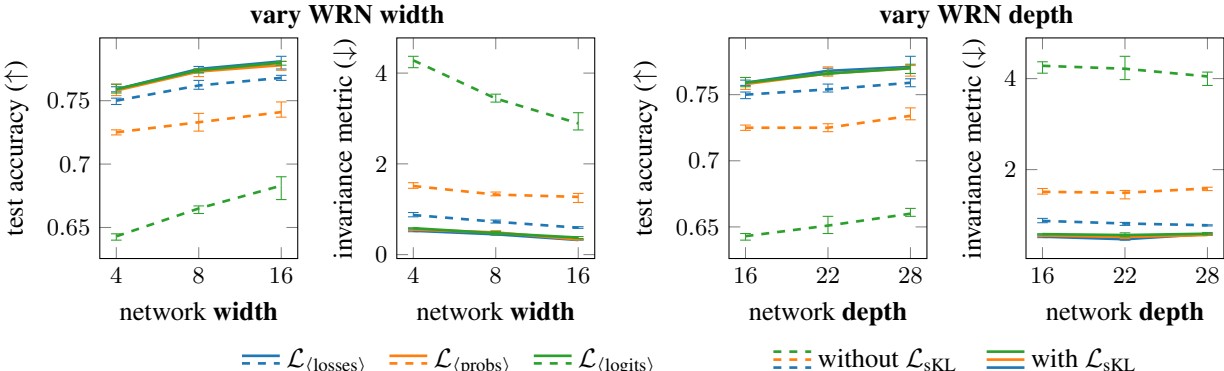

**Figure 6:** WideResNets with varying width factors and depths on CIFAR-100. We train with $K_{\text{train}} = 16$ and evaluate on central crops.

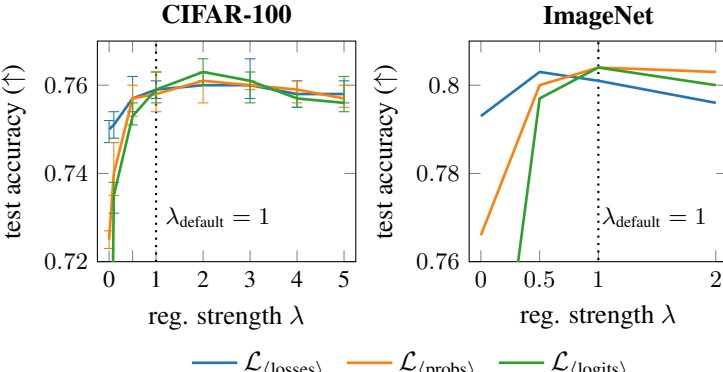

**Figure 7:** Sensitivity of results to the choice of regularisation strength $\lambda$. NF-ResNet-101 with $K_{\text{train}} = 8$ on Imagenet, and WideResNet 16-4 with $K_{\text{train}} = 16$ on CIFAR-100, both evaluated on central crops.

## 5.2 Effect of changing model capacity

In this section, we investigate the effectiveness of the KL regulariser as the capacity of the model changes. For the WideResNet on CIFAR-100 we run additional experiments with varying width and depth. At depth 16 we consider width factors 4 (default), 8, and 16; at width factor 4 we consider depths 16 (default), 22, and 28. For each model we tune the optimal learning rate and optimal epoch budget. From the results in Fig. 6 we can observe that as model capacity increases (both for width and depth), so does the test accuracy of the baselines as expected. More importantly, the models trained with our regulariser improve almost in parallel as well, with the generalisation gap between the two remaining identical. Furthermore, in each case the regulariser also equalises the performance between the three methods. Interestingly, while the invariance measure gets better for wider networks, it stays roughly the same as the network gets deeper.

## 5.3 Sweeping the strength of the KL regulariser

In all our previous experiments, we set the strength of the KL regulariser to $\lambda = 1$. As mentioned in Sec. 4, one important reason why the KL divergence might be well suited in practice as a regulariser is that it is measured in the same units as the log-likelihood, *nats per image*, and hence is on the same scale as the original objective. This suggests that $\lambda \approx 1$ should be a good value for $\lambda$ as it provides equal balance between the original objective and the regulariser.

Here, we provide a comparison when sweeping over the value of $\lambda \in [0, 2 \text{ or } 5]$ for the WideResNet on CIFAR-100 and the NF-ResNet-101 on ImageNet, respectively. The corresponding test accuracies using central-crop evaluation are shown in Fig. 7. We find that the generalisation performance is relatively insensitive to the choice of $\lambda$ for large enough values, and that $\lambda = 1$ consistently yields among the best performance. $\lambda = 0$ corresponds to training without the regulariser and performs markedly worse. Very large values of $\lambda$ also

lead to degraded accuracies and likely correspond to over-regularization. These results suggest that the regulariser weight $\lambda = 1$ is close to optimal, though minor gains are possible through further tuning of $\lambda$.

## 6 Related Work

**Data-augmentation and invariance in neural networks.** The incorporation of invariances is a common inductive bias in machine learning and different approaches have been developed over the years. First, DA methods that enlarge the training set and implicitly enforce a model's predictions to be correct for a larger portion of the input space (Beymer and Poggio, 1995; Niyogi et al., 1998). The augmentations are dataset-dependent and hand-tuned (Cubuk et al., 2020; DeVries and Taylor, 2017; Yun et al., 2019) or sometimes even learned (Cubuk et al., 2019; Wilk et al., 2018). Second, methods that explicitly constrain the intermediate network or its output to be invariant or equivariant to certain transformations. For example, 2D convolutions (LeCun et al., 1989; Lecun et al., 1998) are equivariant to translations while generalised convolutions (Cohen and Welling, 2016) are equivariant to more general group transformations. However it is difficult to hard-code more complex equivariances/invariances. Raj et al. (2017) and Wilk et al. (2018) recently proposed to construct an invariant covariance function for kernel methods by integrating a non-invariant kernel over the augmentation distribution. Nabarro et al. (2021) use this approach to define invariant losses for neural networks by averaging the logits or probabilities as discussed in Sec. 3. Third, regularisation methods that do not place hard constraints on the classifier function or its outputs but instead add an additional loss as a soft-constraint that encourages the desired behaviour. Examples from the self-supervised literature (see next paragraph) and our proposed KL soft-constraint fall into this category. Bouchacourt et al. (2021) recently investigated the implicit effects of DAs on how invariant models are. Though, they focus on sampling a single augmentation per image, whereas we consider training with larger augmentation multiplicities ($K > 1$).

Concurrently to us, other works have explored explicit DA regularisers for empirical risk minimisation. Yang et al. (2022) show that explicitly regularising the features of an augmented image to be similar to the features of the original image can achieve better generalisation than training with empirical risk minimisation on an augmented data set. This approach is similar to the main idea presented in this work, with a few key differences – (i) their regulariser is applied to the features of the penultimate layer, not the predictive distribution, (ii) only the original images images are used in the maximum likelihood part of the objective, and (iii) the regulariser is never applied to two augmented images, but only between the original image and an augmented image. Similarly to us, they also consider multiple augmentation multiplicities, however their experiments are limited to a small subset of CIFAR-100. This approach is also related to the semi-supervised approach of Xie et al. (2020) discussed in the next paragraph. Huang et al. (2021) propose an explicit regulariser that encourages the loss value of the original image and its augmentation to be close. Unlike our regulariser, this does not necessarily enforce features to be similar, as non-equivalent predictions can incur the same loss value. Moreover, they only use single augmentations and focus their analysis on robustness of the predictions to input noise rather than generalisation performance. Wang et al. (2022) empirically study several invariance regularizers for supervised learning with single augmentations. In contrast to their work, we focus on more standard augmentations, consider multiple data augmentations per image, and perform empirical evaluation on the much larger scale ImageNet dataset.

**Self-supervised and contrastive methods** are commonly used to learn visual representations from unlabelled data. Simply speaking, they construct a surrogate "self-" supervised learning problem and use it to train a neural network feature extractor that is subsequently used in other downstream tasks. Because in many methods the features are not grounded (by true labels), additional tricks are typically required to prevent the features from collapsing, such as stopping gradients for some embeddings or using exponentially moving averages of the network weights (Grill et al., 2020; Mitrovic et al., 2021). For example, Doersch et al. (2015), Gidaris et al. (2018), and Noroozi and Favaro (2016) propose hand-crafted tasks such as solving a jigsaw. More recent approaches use DAs to construct surrogate instance discrimination tasks and directly maximise a similarity measure between projected features for different augmentations of the same image (Grill et al., 2020) or solve corresponding clustering problems (Caron et al., 2020). Contrastive methods additionally maximise the discrepancy between augmentations of different images (Chen et al., 2020a,b). In

addition to a contrastive loss, Mitrovic et al. (2021) use a KL regulariser similar to ours; however, where we use the predictive for the true label, they regularise probabilities of a surrogate task. Similar regularisers have also been considered for unlabelled data in semi-supervised settings; for example, Sajjadi et al. (2016) minimise the L2-distance between features, and Xie et al. (2020) propose a cross-entropy-regulariser. Both approaches, however, apply the regularizers only the unsupervised part of the data and have not investigated their effects in the supervised setting. Most methods cited above (with the exception of Mitrovic et al. (2021) and Sajjadi et al. (2016)) only consider pairs of augmentations, whereas we use larger augmentation multiplicities that additionally improve performance as demonstrated by Fort et al. (2021) as well as our experimental results.

## 7 Conclusion

In this paper we investigated implicit and explicit regularisation with data augmentation in supervised learning. We discussed two approaches that both use multiple data augmentations per input but differ in how and at what level they encourage or enforce invariance to data augmentation in the predictor: (i) by averaging the losses of individual augmentations to encourage invariance on the level of the network outputs; or (ii) by averaging the logits or probabilities to make the whole predictor invariant by construction, though network outputs on individual augmentations are not necessarily invariant. We found empirically that the former approach generalises better and that its outputs are more similar across different augmentations of the same image. Motivated by this, we introduced a KL regulariser which explicitly encourages this similarity of the network outputs. Through extensive experiments on CIFAR-100 and ImageNet with multiple large-scale models, we showed that the proposed regulariser improves generalisation performance for all methods and largely equalises performance of the considered approaches. Our results confirm that encouraging invariance on the level of the individual predictions drives the improvements in generalisation performance when using multiple augmentations per image.

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
