# OpenReview forum: "Regularising for invariance to data augmentation improves supervised learning"
_TMLR — Rejected by TMLR_

### Review · Reviewer_mjy6 · 2022-10-11

**Summary Of Contributions:**

The authors perform an empirical study of various ways the objective function can be modified to incorporate data augmentation.
Apart from the standard method of averaging over losses for augmented samples, the authors study Bayesian-inspired logit averaging and probability averaging. Through different experiments, the authors show that loss averaging performs the best; provide a toy example to show why logit averaging does not work and show the relationship between generalization performance and a KL divergence based-invariance metric. Based on insights from this invariance metric, the authors propose to explicitly add this as an auxiliary objective to be optimized and show improvement in performance.

**Requested Changes:**

1. Please improve the formatting of the paper and include legends for the plots.
2. I would like to see experiments with ViTs to see if the conclusions hold.
3. Please perform experiments by chaining augmentations together to see if the conclusions hold.

**Strengths And Weaknesses:**

**Strengths**:
1. Promoting label invariance explicitly seems to be a nice addition to the line of augmentation research which has been done for SSL methods but not so much for Supervised methods.
2. The invariance metric provides a glimpse into how different augmentations alter the logit distribution.

**Weaknesses**:
1. The formatting of the paper can be substantially improved. I am not sure if this is only a bug for me but none of the plots have legends to explain what the different lines correspond to. This makes interpreting the results extremely time-consuming and difficult.
2. AutoAugment (Cubuk et.al) has shown that chaining augmentations also improves performance, while this work mainly focuses on single augmentation at once. I would like to see if the results hold for multiple augmentations.
3. Powerful augmentations like CutMix, StyleMix, MixUp are not studied.
4. The model families considered are extremely limited. For an empirical conclusion, It would be good to see if these results generalize to ViTs which are becoming popular by the day.
5. The authors mention they are using Jeffrey's divergence instead of KL Divergence yet they never define it or use a different symbol for it which makes it confusing for the reader.
6. Is there a reason why the authors do not minimize the KL divergence with the original unaugmented sample for the invariance metric and only perform it amongst the augmented inputs?
7. Some of the references seem incomplete (eg. Yang et al.)
8. Some preliminary results on models with batch normalization would be interesting to see.
** Other Questions**:
1. Does it make sense to combine $\mathcal{L}_{loss}$ and $\mathcal{L}_{logits}$ together for better performance?

---

### Review · Reviewer_nEpA · 2022-10-14

**Summary Of Contributions:**

This paper investigates the relationship between data augmentation and invariance in supervised learning, it is very interesting to see the discussion of the three ways of leveraging the augmentation data, which is very rarely discussed in the literature in my opinion. However, on the other hand, the proposed regularization method has been discussed in previous literature in a much more extensive manner.

**Broader Impact Concerns:**

none noted.

**Requested Changes:**

I will recommend the authors to follow the experiment settings in [1], e.g.,
 - proposing a proper definition of invariance, not necessarily the same as the ones in [1], but at least more formally designed, and generic enough to be measured by different divergences or distance metrics.
 - comparing the results in a way that the proposed method is evaluated in a way that is different from the evaluation metric (one cannot use KL (or its minor variants) in both cases)
 - clearly contrast the differences in comparison to [1], maybe even with a dedicated paragraph since the second half of the contribution of this paper is too close to the overall theme of [1].

**Strengths And Weaknesses:**

- Strength
  - interesting discussion on the three ways of using augmented data. The idea seems quite straightforward and somehow rarely discussed in other literature.
  - The paper is written with great clarity, the way of grouping eqs 5, 6, 7 makes the reader easily appreciate the discussions.

- Weakness
  - missing significant references, for example, the second half of the contributions (the regularized method) does not seemingly add any contributions upon [1].
  - the invariance metric depends on a pre-chosen metric (KL), thus, the discussion might be different if we use a different metirc, especially since KL is asymmetric (later the authors uses the symmetric version of it, but it seems the evaluation still uses the asymmetric version? Regardlessly, a different metric, like Euclidean distance might change the conclusion).
  - there seems a significant flaw in experiment design: if the regularizing method is KL (or its symmetric version), the results evaluated by KL will certainly favor it.


[1]. Toward Learning Robust and Invariant Representations with Alignment Regularization and Data Augmentation, KDD 2022

---

### Review · Reviewer_hANv · 2022-11-03

**Summary Of Contributions:**

This paper proposes an invariance regularization term with multiple data augmentation to enhance the generalization of a supervised learning model. The authors first perform an empirical study of various invariance regularization techniques, and find out that the averaging the loss of individual augmentations to enforce invariance at the network output is more effective than others. Based on this observation, the authors propose a KL regularizer which explicitly encourages the similarity in the network output, and experimentally validate its performance on benchmark datasets. The experimental results confirms the efficacy of the proposed regularization term.


**Requested Changes:**

None

**Strengths And Weaknesses:**

Pros
- The proposed KL divergence-based regularizer on the network output is well-motivated, backed by a thorough empirical study of various invariance-based regularization techniques.
- The empirical study of different explicit and implicit invariance regularization techniques at different levels (losses, probs, and logits) itself is interesting and provides some useful insights to the practitioners.
- The proposed method seems effective in enhancing the performance of a supervised model.
- The paper is well-written in general with clear descriptions of the methods, well-presented experimental results, polished editing, and more than adequate references.

Cons
- As mentioned by the authors themselves, similar consistency regularization at the network output level was exploited for self-supervised learning and semi-supervised learning, which limits the novelty of the proposed work to the application of the regularizer to a supervised learning setting. Therefore, it is questionable if this is a sufficient novelty or contribution to warrant publication, although TMLR does not emphasize on the novelty.

---

### Author Response · Authors · 2022-10-14
**Updated PDF to fix legends**

Thank you very much for pointing out the broken legends in all figures. We are very sorry for this and the inconvenience caused.

We've now updated the PDF to fix the legends.

---

### Decision · Action_Editors · 2023-01-03

**Recommendation:** Reject

**Comment:**

Both claims are clearly stated and well-motivated empirically. However, as pointed out by all three reviewers, a similar consistency regularizer at the network output level was proposed before. Therefore, the novelty is drastically limited and, as a result, it becomes questionable if the presented experiments could be seen as a sufficient contribution. In view of the two claims, this puts in question the second claim of the paper. As a result, I do not see the first claim to be sufficient for accepting the paper as it is. Please find a list of questions raised by the reviewers below, since they could be of great help in improving your paper in the future.

1. Please improve the formatting of the paper and include legends for the plots.
2. Please include experiments with ViTs to see if the conclusions hold.
3. Please perform experiments by chaining augmentations together to see if the conclusions hold.
4. Please provide a proper definition of invariance, not necessarily the same as the ones in [1], but at least more formally designed, and generic enough to be measured by different divergences or distance metrics.
5. Please compare the results in a way that the proposed method is evaluated differently from the evaluation metric (one cannot use KL (or its minor variants) in both cases).
6. Please contrast the differences in comparison to [1], maybe even with a dedicated paragraph since the second half of the contribution of this paper is too close to the overall theme of [1].

[1]. Toward Learning Robust and Invariant Representations with Alignment Regularization and Data Augmentation, KDD 2022

**Audience:**

The paper attacks a fundamental problem in machine learning. As a result, it is of great interest to the TMLR audience.

**Claims And Evidence:**

In this paper, a new invariance regularization term with multiple data augmentation is proposed. First, the authors first carry out experiments for various invariance regularizers. The results suggest that averaging the loss of individual augmentations to enforce invariance at the network output is more effective than others. Next, the authors propose a Kullback-Leibler-based regularizer which explicitly encourages the similarity in the network output. Their proposition is further validated experimentally. The experimental results confirm the efficacy of the proposed regularizer.

The paper makes two claims:
1. Averaging the losses for explicit and implicit invariance outperforms Bayesian-inspired losses and results in more invariant model predictions.
2. A new regularizer for supervised learning tasks is proposed that explicitly encourages predictions. Further, it is shown that this new regularizer outperforms all base losses.

The claims are clearly stated. The paper is well-organized and easy to follow. However, all three reviewers agreed that even though the paper is rather strong empirically, it requires more comparisons and more in-depth analysis. Moreover, the newly proposed regularizer is not new, as also mentioned by the authors.